# Human WDR5 promotes breast cancer growth and metastasis via KMT2-independent translation regulation

Wesley L Cai[1,2†], Jocelyn Fang-Yi Chen[2†], Huacui Chen[2], Emily Wingrove[2], Sarah J Kurley[3], Lok Hei Chan[2], Meiling Zhang[2], Anna Arnal-Estape[2,4], Minghui Zhao[2], Amer Balabaki[2], Wenxue Li[5], Xufen Yu[6,7], Ethan D Krop[2,8], Yali Dou[9,10], Yansheng Liu[4,5], Jian Jin[6,7], Thomas F Westbrook[3], Don X Nguyen[2,4,11,12]*, Qin Yan[2,4,11,13]*

[1]Hillman Cancer Center, University of Pittsburgh Medical Center, Pittsburgh, United States; [2]Department of Pathology, Yale University, New Haven, United States; [3]Department of Biochemistry and Molecular Biology, Baylor College of Medicine, Houston, United States; [4]Yale Cancer Center, Yale School of Medicine, New Haven, United States; [5]Yale Cancer Biology Institute, Department of Pharmacology, Yale University, West Haven, United States; [6]Mount Sinai Center for Therapeutics Discovery, Icahn School of Medicine at Mount Sinai, New York, United States; [7]Departments of Pharmacological Sciences and Oncological Sciences, Tisch Cancer Institute, Icahn School of Medicine at Mount Sinai, New York, United States; [8]Department of Biosciences, Rice University,, Houston, United States; [9]Department of Pathology, University of Michigan, Ann Arbor, Ann Arbor, United States; [10]Department of Medicine, Department of Biochemistry and Molecular Medicine, University of Southern California, Los Angeles, United States; [11]Yale Stem Cell Center, Yale School of Medicine, New Haven, United States; [12]Department of Internal Medicine (Section of Medical Oncology), Yale School of Medicine,, New Haven, United States; [13]Yale Center for Immuno-Oncology, Yale School of Medicine, New Haven, United States

*For correspondence:
don.nguyen@yale.edu (DXN);
qin.yan@yale.edu (QY)

†These authors contributed
equally to this work

**Abstract** Metastatic breast cancer remains a major cause of cancer-related deaths in women, and there are few effective therapies against this advanced disease. Emerging evidence suggests that key steps of tumor progression and metastasis are controlled by reversible epigenetic mechanisms. Using an in vivo genetic screen, we identified WDR5 as an actionable epigenetic regulator that is required for metastatic progression in models of triple-negative breast cancer. We found that knockdown of WDR5 in breast cancer cells independently impaired their tumorigenic as well as metastatic capabilities. Mechanistically, WDR5 promotes cell growth by increasing ribosomal gene expression and translation efficiency in a KMT2-independent manner. Consistently, pharmacological inhibition or degradation of WDR5 impedes cellular translation rate and the clonogenic ability of breast cancer cells. Furthermore, a combination of WDR5 targeting with mTOR inhibitors leads to potent suppression of translation and proliferation of breast cancer cells. These results reveal novel therapeutic strategies to treat metastatic breast cancer.

## Editor's evaluation

Using combined in vivo and in vitro screens, this study identifies WDR5 as important for tumour growth and lung metastasis in a triple negative breast cancer. WDR5 promotes global translation

rates and enhanced ribosomal protein expression, and targeting of WDR5 in combination with MTOR inhibition effectively reduces tumour cell growth and metastasis. Novel therapeutic strategies for triple negative breast cancer are urgently needed, and this study elegantly provides a novel therapeutic strategy that may contribute to improved clinical management of this patient population.

## Introduction

In the United States, metastatic breast cancer is the second leading cause of cancer-related death among women (*Harbeck et al., 2019*; *Torre et al., 2017*). In particular, triple-negative breast cancer (TNBC) has the worst prognosis among all breast cancer subtypes, largely owing to its high metastatic proclivity to the lungs and other sites and because there are few effective treatments against this disease once it has metastasized (*Al-Mahmood et al., 2018*). Recently developed targeted therapies for TNBC, including poly (ADP-ribose) polymerase inhibitors or immune checkpoint inhibitors, are effective in patients whose tumors express BRCA1/2 mutations or high programmed death-ligand 1, respectively (*Gonzalez-Angulo et al., 2011*; *Lyons and Traina, 2019*). However, these patients account for only 9.3–15.4% of TNBC cases (*Armstrong et al., 2019*), and new treatment strategies are urgently needed.

Emerging evidence suggests that tumor growth is modulated by reversible epigenetic mechanisms (*Blair and Yan, 2012*; *Cao et al., 2014*; *Cao and Yan, 2013*; *Chen and Yan, 2021*). In primary human breast cancers, we and others recently identified distinct chromatin states, which distinguish established molecular subtypes and correlate with metastatic relapse and poor clinical outcome (*Cai et al., 2020*). Therefore, regulators of histone modifications and chromatin dynamics, in particular, may be required for breast cancer progression. The identity of such regulators as well as strategies to therapeutically target them in the metastatic setting remain unclear.

Epigenetic regulators that are known to be involved in tumorigenesis include the KMT2 (also known as MLL/SET1) family protein complexes which mark active promoters and enhancers with histone H3 lysine K4 (H3K4) methylation and the non-specific lethal (NSL) complex which acetylates histones (*Dias et al., 2014*; *Raja et al., 2010*; *Ruthenburg et al., 2007*; *Wysocka et al., 2005*). WDR5 is a WD40 repeat protein that is a scaffold for the assembly of the KMT2 and NSL complexes (*Guarnaccia and Tansey, 2018*). More recently, WDR5 was found to physically interact with the proto-oncogene and transcription factor MYC to guide its chromatin binding and transcriptional activation, suggesting that WDR5 is a tractable target for MYC-driven cancers (*Thomas et al., 2015a*; *Thomas et al., 2015b*). Aberrant WDR5 expression itself may occur in a number of cancer types (*Chen et al., 2015*; *Dai et al., 2015*; *Ge et al., 2016*). Biologically, WDR5 may contribute to tumor sphere formation and cell proliferation (*Carugo et al., 2016*; *Chung et al., 2016*). Molecularly, WDR5 is reported to modulate the expression of various genes, which may be specific to cell type or cell state (*Bryan et al., 2020*; *Oh et al., 2020*). On the other hand, WDR5 was recently discovered to broadly regulate the expression of ribosomal protein (RP) genes across multiple cell lines and cancer types (*Aho et al., 2019*; *Bryan et al., 2020*; *Guarnaccia et al., 2021*). Moreover, deregulation of RP gene expression and translation has been implicated in breast cancer metastasis (*Ebright et al., 2020*). However, the relative importance of different WDR5 effector functions and their requirement for breast cancer progression and metastasis have not been well studied.

Here, we established an in vivo screening platform that identified WDR5 as a key regulator of breast cancer cell growth and metastatic colonization. We further showed that WDR5 regulates RP gene expression and global protein translation independently of the KMT2 complex. Moreover, our results indicate that WDR5 inhibition or degradation could be used as a therapeutic approach for TNBC and that WDR5 targeting could be combined with mTOR inhibitors to achieve significant therapeutic benefit.

## Results

### The establishment of in vivo lung metastasis screening platform

To identify actionable epigenetic targets for breast cancer metastasis, we conducted parallel in vivo and in vitro functional screens using an inducible, barcoded short hairpin RNA (shRNA) library (*Figure 1A*). We first compiled a list of epigenetic regulators based on: (1) if they could be targeted

**Figure 1.** Decreasing WDR5 reduces breast cancer cell growth and lung metastasis. (**A**) Schematic of in vivo and in vitro screening work flow. Epigenetic regulator inducible knockdown cell lines were mixed in equal numbers (pool of 8–10 lines) with control cell lines and injected into control or doxycycline (DOX) treated mice (625 mg/kg) intravenously or cultured under control or DOX (1 μg/mL) treated condition. Both lungs (in vivo) and cells (in vitro) were harvested for gDNA and subjected to barcode quantitative PCR as the screening output. (**B**) Volcano plot showing the results of the in vivo screen. Each data point is an average of 10–20 mice. Discovery hits are selected using p<0.05 and log2FC (fold change)>0.8 or log2FC <−0.8. (**C**) Log2FC of the in vivo screen results versus log2FC of the in vitro screen results for each of the epigenetic regulators. Discovery hits are

*Figure 1 continued on next page*

*Figure 1 continued*

selected as in (**B**). (**D**) Western blot analysis of the indicated proteins in LM2 cells harboring inducible control (shCtrl) or WDR5 targeting (shWDR5-1 and shWDR5-2) shRNA after 3 days of DOX induction. (**E**) WST-1 proliferation assays of LM2 cells from (**D**) after indicated days of DOX treatment. Each symbol indicates mean ± SD for representative experiment performed in quadruplicate (n=4, unpaired two-side Student's *t* test). ****p<0.0001 (shCtrl versus shWDR5-1) and ****p<0.0001 (shCtrl versus shWDR5-2). (**F–G**) Colony formation assays of LM2 cells from (**D**) after 9 days of either control or DOX treatment. Representative images (**F**) and quantification (**G**) are shown (n=3, unpaired two-side Student's *t* test). p=0.057 (shCtrl − DOX versus shCtrl +DOX), ****p<0.0001 (shWDR5-1 −DOX versus shWDR5−1 +DOX), and **p=0.0025 (shWDR5-2 −DOX versus shWDR5−2 +DOX). (**H**) Normalized bioluminescence unit (BLU) signals of lung metastasis of mice injected intravenously with LM2 cells from (**D**) and kept under DOX chow. The data represent mean ± SEM (shCtrl: n=6; shWDR5-1: n=7; shWDR5-2: n=5). **p=0.0012 (shCtrl versus shWDR5-1) and *p=0.017 (shCtrl versus shWDR5-2). (**I–J**) Box plots of relative BLU of indicated cell line at day 7 (**I**), p=0.1375 (shCtrl versus shWDR5-1), *p=0.0173 (shCtrl versus shWDR5-2), and day 50 (**J**), **p=0.0012 (shCtrl versus shWDR5-1), *p=0.017 (shCtrl versus shWDR5-2), post-injection normalized to its day 0 value. (**K**) Tumor volume measurements of mice injected into the fourth mammary fat pad with LM2 cells harboring inducible control or shWDR5-1. The data represent mean ± SEM. ****p<0.0001 (shCtrl versus shWDR5-1). (**L**) Representative BLU images of mice in (**K**) at day 61. Significance determined using unpaired two-tailed Mann–Whitney test (shCtrl: n=14; shWDR5-1: n=13). *p<0.05; **p<0.01; ***p<0.001; ****p<0.0001. For gel source data, see *Figure 1—source data 1*.

The online version of this article includes the following source data and figure supplement(s) for figure 1:

**Source data 1.** Original western blots for *Figure 1D*.

**Figure supplement 1.** In vitro and in vivo distribution of positive control shRNAs and negative control shRNAs in pooled screens.

with existing pharmacological agents, (2) if their expression correlated with poor survival in multiple independent datasets (hazard ratio >1 and p-value<0.05), or (3) if their expression was increased in the lung metastatic cell subpopulation MDA-MB-231-LM2 (LM2) cells when compared to the parental TNBC cell line MDA-MB-231. We designed our screen using the LM2 cells because they reproducibly generate lung metastasis, and lung is the most frequent site of distant relapse in TNBC patients (*Lin et al., 2008*; *Minn et al., 2005*). Accordingly, we tested the knockdown efficiency of 327 shRNAs targeting 89 epigenetic regulators and selected one shRNA with the best knockdown efficiency per target gene (*Supplementary file 1a*). We then subcloned these 89 shRNAs into the doxycycline (DOX) inducible and barcoded pINDUCER10 lentivirus to generate a focused knockdown-validated shRNA library (*Figure 1—figure supplement 1A*; *Meerbrey et al., 2011*). Included in this library were positive control shRNAs against *BUD31* (shBUD31) and *SAE2* (shSAE2), which were previously shown to be essential for LM2 cell proliferation, along with shRNAs against *CHEK1* and *STAMBP*, which served as negative controls (*Hsu et al., 2015*; *Kessler et al., 2012*).

LM2 cells were infected with individual shRNAs targeting 64 epigenetic regulators and control shRNA from this library. To ensure that any particular hairpin had enough representation and was above the detection limit for in vivo screening, a subset of knockdown cell lines against 8–10 epigenetic regulators and the control cell lines were combined together in equal numbers into seven minipools. We cultured these minipools in either control or DOX conditions for up to 10 doublings, then extracted gDNA from the pooled shRNA infected cells collected. Quantitative PCR (qPCR) analysis of gDNA confirmed that shBUD31 and shSAE2 were significantly depleted in the DOX-treated pools (*Figure 1—figure supplement 1B*). On the other hand, the amount of shSTAMBP and shCHEK1 expressing cells did not change significantly in most control or DOX conditions, while shCHEK1 cells increased slightly after 10 days of DOX treatment (*Figure 1—figure supplement 1C*). We next determined whether our controls perform similarly in vivo. Minipools were then injected intravenously and treated with either control or DOX (in animal chow for in vivo conditions). After 50 days, tumor-bearing lung tissue was collected and processed for gDNA extraction. We then compared the barcode abundance between control and DOX-treated lung tissue using qPCR analysis of gDNA. The results were normalized to the day 0 value.

We found that DOX treatment does not in itself affect the in vivo kinetics of lung metastatic growth (*Figure 1—figure supplement 1D*). In a representative minipool, shBUD31 and shSAE2 consistently dropped-out in the DOX-treated condition, whereas shSTAMBP remained unchanged (*Figure 1—figure supplement 1E*). shCHEK1 was enriched significantly, which is likely indirectly due to the depletion of other shRNA expressing cell lines in the minipools (*Figure 1—figure supplement 1E*). From the in vivo screen, we identified 14 significant hits (p<0.05, $\log_2$FC >0.8 or $\log_2$FC <−0.8, FC:+DOX/−DOX), and among these, 7 were drop-out hits where shRNA representation significantly decreased, while 7 were enrichment hits where shRNA representation significantly increased (*Figure 1B*, *Supplementary file 1b*). Many of these in vivo drop-out candidates were drop-out candidates in vitro (*Figure 1C*,

*Supplementary file 1b*). For example, our screen identified drop-out shRNAs against MCM6, an essential eukaryotic genome replication factor, and CSNK2A1, previously shown to enhance metastatic growth of MDA-MB-231 cells (*Bae et al., 2016*; *Figure 1B and C*). Thus, many of these epigenetic targets may be, at least in part, required for the cell intrinsic fitness of metastatic cells.

## Decreasing WDR5 reduces breast cancer cell growth and lung metastasis

Among the top hits and potential therapeutic targets that were identified, we focused on WDR5 because it can be inhibited by small molecules, and shWDR5 was the second most significantly depleted shRNA in vivo (after shMCM6). WDR5 is known canonically as a scaffolding protein that recognizes and binds to methylated H3K4, allowing the modification of H3K4 trimethylation by the KMT2 protein complex (*Wysocka et al., 2005*). More recently, WDR5 has been discovered to physically interact with and guide MYC to its transcriptional targets (*Thomas et al., 2015b*). WDR5 has also been implicated in the growth of metastatic breast cancer cells, although the mechanism underlying this function of WDR5 is not clearly delineated (*Punzi et al., 2019*).

We first confirmed the knockdown effect of WDR5 in individual unpooled LM2 cells and by using independent shRNAs against WDR5 (shWDR5-1 and shWDR5-2). Following 3 days of shRNA induction in vitro (*Figure 1D*), both shRNAs against WDR5 caused a significant but modest decrease in cell proliferation when compared to a control shRNA (shCtrl) over 5 days (*Figure 1E*). In addition, using long-term in vitro colony formation assays over 9 days, we found a profound impact of WDR5 knockdown on the in vitro clonogenic ability of LM2 cells (*Figure 1F and G*). We next asked whether both shRNAs affect lung metastasis outgrowth in vivo. We induced knockdown for 3 days in vitro before injecting LM2 cells into the tail vein of mice and monitoring lung metastatic colonization and outgrowth over 50 days. We observed a significant impairment on lung colonization by LM2 cells as early as day 7 post-injection (*Figure 1H and I*). At the end point (day 50), the average lung metastatic burden in the mice with shWDR5 cells was 5.7- or 16.5-fold lower than that in mice with LM2 cells expressing the control shRNA (*Figure 1H and J*). In addition to metastatic colonization from circulation, we tested whether knockdown of WDR5 affects tumor growth and metastasis from the orthotopic mammary fat pad. We observed a significant decrease in mammary tumor growth in the shWDR5 group compared to control tumors (*Figure 1K and L*, *Figure 1—figure supplement 1F-H*). Notably, we observed an even larger decrease in lung and liver metastasis from the mammary fat pad tumors in the shWDR5 group as compared to shCtrl, which suggests a potential metastasis-specific function of WDR5 (compare *Figure 1—figure supplement 1I-J* to *Figure 1—figure supplement 1H*). Taken together, we showed that WDR5 is independently required for the cellular outgrowth, tumorigenic, and lung colonizing capacities of LM2 TNBC cells.

## WDR5 depletion significantly reduces breast cancer cell growth across multiple breast cancer subtypes

Next, we tested the requirement for WDR5 in other cell line models and from distinct breast cancer subtypes. To this end, we reduced WDR5 in additional breast cancer lines spanning three established molecular subtypes: TNBC (MDA-MB-453 and HCC1143), estrogen receptor positive (MCF7, T47D, and MDA-MB-361), and HER2+ (UACC893, BT474, and SKBR3) (*Figure 2A*). WDR5 knockdown significantly reduced the clonogenic outgrowth of all the tested cell lines (*Figure 2B and C*), suggesting that WDR5 enhances tumor cell growth across multiple breast cancer subtypes. As we were particularly interested in evaluating the therapeutic potential of targeting WDR5 in TNBC, we next tested the efficacy of a known WDR5 inhibitor, OICR-9429, which is a small molecule antagonist of the WDR5-KMT2 interaction (*Grebien et al., 2015*). We treated LM2 cells with OICR-9429 at 20 μM for 9 days, and this also significantly reduced their colony formation ability (*Figure 2D and E*). Similar results were found when using OICR-9429 to treat two other TNBC cell lines, MDA-MB-453 and 4T1, although we noted that growth inhibition was more significant in MDA-MB-453 cells when using 30 μM of OICR-9429 (*Figure 2F and G*, *Figure 2—figure supplement 1A and B*).

As the effective concentration of OICR-9429 is relatively high and may lead to off-target effects, we sought to test the effect of our recently published WDR5 degrader MS67, which recruits WDR5 to Cullin4-CRBN E3 ubiquitin ligase complex for proteasome-mediated degradation (*Yu et al., 2021*). We first evaluated the effect of MS67 on degrading WDR5 in LM2 and MDA-MB-453 cells. We found that



**Figure 2.** WDR5 targeting significantly reduces breast cancer cell growth across breast cancer subtypes. (**A**) Western blot analyses of WDR5 in the indicated cell lines infected with either control or WDR5-targeting hairpins with or without 3 days of doxycycline (DOX) (1 µg/mL) induction. (**B**–**C**) Colony formation assays of indicated control or shWDR5-1 cell lines from (**A**) after 9 days of DOX (1 µg/mL) treatment. Representative images (**B**) and quantification (**C**) are shown (n=3, unpaired two-side Student's *t* test). **$p_{MDA-MB-453}$=0.0011, ****$p_{HCC1143}$ <0.0001, *$p_{MCF7}$=0.0263, ****$p_{T47D}$<0.0001, ****$p_{MDA-MB-361}$<0.0001, **$p_{UACC893}$=0.0017, ***$p_{BT474}$=0.0004, and *$p_{SKBR3}$=0.0156. Cell lines are grouped by breast cancer molecular subtype. (**D**–**G**) Colony formation assays of LM2 (**D**) and MDA-MB-453 (**F**) after 9 days of either control or OICR-9429 treatment at the indicated concentration. Representative images (D

*Figure 2 continued on next page*

*Figure 2 continued*

and F) and quantification (E and G) are shown (n=3, unpaired two-side Student's *t* test). ****p$_{LM2}$ <0.0001 (dimethyl sulfoxide (DMSO) versus OICR-9429 20 µM), **p$_{MDA-MB-453}$=0.0018 (DMSO versus OICR-9429 20 µM), and ****p$_{MDA-MB-453}$<0.0001 (DMSO versus OICR-9429 30 µM). (**H**) Western blot analysis of WDR5 in LM2 cells treated with MS67, MS67N, or OICR-9429 at the indicated concentrations for 18 hr. Band intensities of WDR5 were quantified by image J and normalized by those of vinculin control. (**I–J**) Colony formation assays of LM2 after 9 days of treatment with control, OICR-9429, MS67N, or MS67 at the indicated concentrations. Representative images (**I**) and quantification (**J**) are shown (n=3, unpaired two-side Student's *t* test). **p$_{0.5µM}$=0.0012 (MS67N versus MS67), **p$_{2.5µM}$ = 0.0033 (DMSO versus OICR-9429), and ****p$_{2.5µM}$ <0.0001 (MS67N versus MS67). For gel source data, see *Figure 2—source data 1* and *Figure 2—source data 2*.

The online version of this article includes the following source data and figure supplement(s) for figure 2:

**Source data 1.** Original western blots for *Figure 2A*.

**Source data 2.** Original western blots for *Figure 2H*.

**Figure supplement 1.** WDR5 inhibition and MS67-mediated WDR5 degradation significantly reduces breast cancer cell growth.

**Figure supplement 1—source data 1.** Original western blots for *Figure 2—figure supplement 1C*.

**Figure supplement 1—source data 2.** Original western blots for *Figure 2—figure supplement 1D*.

**Figure supplement 1—source data 3.** Original western blots for *Figure 2—figure supplement 1E*.

MS67, but not the negative control MS67N, which does not bind to CRBN, nor OICR-9429, induced WDR5 degradation at a concentration as low as 0.02 µM (*Figure 2H*, *Figure 2—figure supplement 1C*). Specifically, at 2.5 µM MS67, we achieved ~80% WDR5 degradation in LM2 cells and ~70% of degradation in MDA-MB-453 cells (*Figure 2H*, *Figure 2—figure supplement 1C*). Additionally, the maximal degradation can be achieved at 8 hr post-treatment, and this effect remains stable for 72 hr in both LM2 and MDA-MB-453 cells (*Figure 2—figure supplement 1D and E*). Finally, we compared MS67-induced WDR5 degradation to OICR-9429 treatment on the clonogenic outgrowth of LM2 and MDA-MB-453 cells. We found that MS67 leads to ~50% growth inhibition at 0.5 µM and ~80% inhibition at 2.5 µM (*Figure 2I and J*, *Figure 2—figure supplement 1F and G*). Importantly, the effect of 2.5 µM MS67 treatment is comparable to shRNA knockdown and more potent than 20 µM OICR-9429 treatment in LM2 and MDA-MB-453 cells, while 2.5 µM of OICR-9429 treatment only caused a modest effect (compare *Figures 1F, G–2B, 2C, I and J*, *Figure 2—figure supplement 1F and G*). In summary, MS67-mediated WDR5 degradation showed improved growth inhibition of breast cancer cells when compared to the OICR-9429 compound.

## WDR5 targeting decreases RP gene expression and global translation rates

To identify the molecular effects of WDR5 depletion in TNBC cells, we performed transcriptomic profiling of control and WDR5 knockdown cells. In addition to the MDA-MB-231 lung metastatic LM2 cells, we also tested the effect of WDR5 knockdown in independent MDA-MB-231 subpopulations that metastasize more readily to the brain (BrM3) or bone (BoM) (*Bos et al., 2009*; *Kang et al., 2003*). Because WDR5 has previously been shown to facilitate active transcription (*Ang et al., 2011*; *Wysocka et al., 2005*), we used spike-in RNA for normalization and found no changes in global RNA levels 3 days after shWDR5-1 induction (*Jiang et al., 2011*). Our analyses identified differentially expressed genes (DEGs) in all three organotropic-metastatic cell subpopulations (*Figure 3A*). In general, inhibition of WDR5 led to more down-regulated genes than up-regulated genes, which supports previous findings that WDR5 generally promotes transcriptional activation (*Wysocka et al., 2005*; *Figure 3A*). Certain DEGs were preferentially regulated in LM2, BrM3, or BoM cells, suggesting that WDR5 can regulate genes in a manner that is dependent on the metastatic proclivities of different breast cancer cell subpopulations (*Figure 3B*). On the other hand, we also found DEGs (264 down-regulated and 118 up-regulated) that were shared across all lines (*Figure 3B*, *Supplementary file 1c*), indicative of some conserved WDR5 function across metastatic breast cancer cells.

We then performed Enrichr analysis on both the up- or down-regulated DEGs and found that the most enriched and significant gene ontology in the shared down-regulated DEGs was cytoplasmic RPs (*Figure 3C*, *Figure 3—figure supplement 1A*, *Supplementary file 1d*). The combined score for this enrichment was 18-fold higher than the next enriched ontology for the shared down-regulated DEGs, demonstrating the significance of this WDR5 regulated pathway (*Figure 3C*, *Supplementary file 1d*). Notably, among the 474 down-regulated DEGs in LM2 cells, 51 (11%) encoded for RP

**Figure 3.** WDR5 targeting decreases ribosomal protein (RP) gene expression and global translation rates. (**A**) Bar graph summarizing the number of differentially expressed genes (DEGs) after WDR5 silencing across three MDA-MB-231 organotropic sublines (LM2-lung; BrM3-brain; BoM-bone). (**B**) Venn diagram showing the number of overlap or distinct down-regulated genes (left) and up-regulated genes (right) after WDR5 knockdown in the MDA-MB-231 organotropic sublines. (**C**) Gene ontology results using the down-regulated gene set shared by all three MDA-MB-231 organotropic sublines analyzed with Enrichr. (**D**) Volcano plot of DEGs after WDR5 knockdown in LM2. Shared DEGs across all lines highlighted in dark red and RP genes highlighted in light red. The top 10 differentially expressed RPs are labeled. (**E**) RT-quantitative PCR (qPCR) validation of selected DEGs in LM2

*Figure 3 continued on next page*

*Figure 3 continued*

cells harboring shCtrl, shWDR5-1, or shWDR5-2 after doxycycline (DOX) (1 μg/mL) induction for 3 days. For *WDR5*, ****p<0.0001 (shWDR5-1 −DOX versus +DOX) and ***p=0.0002 (shWDR5-2 −DOX versus +DOX); for *RPL7*, **p=0.0014 (shWDR5-1 −DOX versus +DOX), and **p=0.0012 (shWDR5-2 −DOX versus +DOX); for *RPL9*, *p=0.0106 (shWDR5-1 −DOX versus +DOX) and **p=0.0024 (shWDR5-2 -DOX versus +DOX); for *RPL31*, ***p=0.0007 (shWDR5-1 −DOX versus +DOX) and ****p<0.0001 (shWDR5-2 −DOX versus +DOX); for *RPL32*, **p=0.0016 (shWDR5-1 −DOX versus +DOX) and **p=0.0018 (shWDR5-2 −DOX versus +DOX); for *RPS14*, **p=0.0015 (shWDR5-1 −DOX versus +DOX) and **p=0.0084 (shWDR5-2 −DOX versus +DOX). (**F**) RT-qPCR validation of selected DEGs in LM2 cells after DMSO or OICR-9429 treatments at the indicated concentration for 3 days. For *RPL7*, **p=0.0068 (DMSO versus OICR-9429 10 μM) and *p=0.0138 (DMSO versus OICR-9429 20 μM); for *RPL32*, *p0.0268 (DMSO versus OICR-9429 10 μM) and *p=0.0106 (DMSO versus OICR-9429 20 μM); for *RPS14*, **p=0.0032 (DMSO versus OICR-9429 10 μM) and **p=0.0024 (DMSO versus OICR-9429 20 μM). (**G**) RT-qPCR validation of selected DEGs in LM2 cells after DMSO, OICR-9429, MS67N, or MS67 treatments at the indicated concentration for 48 hr. Significance determined by comparing each treatment to DMSO control (n=4, unpaired two-side Student's *t* test). For *RPL32*, *p=0.0171 (DMSO versus OICR-9429 2.5 μM), **p=0.0056 (DMSO versus MS67 0.5 μM), and **p=0.0031 (DMSO versus MS67 2.5 μM); for *RPS14*, **p=0.0068 (DMSO versus MS67N 0.5 μM), *p=0.0303 (DMSO versus MS67N 2.5 μM), and **p=0.0033 (DMSO versus MS67 2.5 μM). (**H–J**) Normalized translation rates as measured by incorporation of methionine analog homopropargylglycine (HPG) over time and evaluated by flow cytometry. Each data point represents the slope of HPG incorporation for at least three time points using median fluorescence intensity from an independent experiment. LM2 cells from (**E**) following 3 days of DOX (1 μg/mL) induction (**H**), ****$p_{shWDR5-1}$<0.0001, *$p_{shWDR5-2}$=0.0230, LM2 cells following 3 days of control or OICR-9429 treatment at 20 μM (**I**), **p=0.0043, and LM2 cells following 3 days of MS67N or MS67 treatment at 2.5 μM (**J**), *p=0.0128, were tested. (n=3, one sample *t* test). *p<0.05; **p<0.01; ***p<0.001; ****p<0.0001.

The online version of this article includes the following figure supplement(s) for figure 3:

**Figure supplement 1.** WDR5 targeting decreases ribosomal protein (RP) gene expression and global translation rates.

genes (*Figure 3D*). A similar enrichment pattern was observed in BrM3 and BoM cells (*Figure 3—figure supplement 1B and C*). After inducing WDR5 knockdown with both hairpins for 3 days in LM2 cells, we confirmed down-regulation of all the tested RP genes. These included the top two down-regulated RPs, *RPL7* and *RPL31*, which were consistently reduced by a ~50% (*Figure 3E*). We next tested whether WDR5 targeting with either OICR-9429 or MS67 would have similar effects on gene expression. Treatment of LM2 and MDA-MB-453 cells with 10 or 20 μM of OICR-9429 for 3 days decreased *RPL7* and other RP genes as predicted (*Figure 3F*, *Figure 3—figure supplement 1D*). We next evaluated the gene-regulatory effect of MS67, which is a more effective inhibitor of WDR5. MS67 treatment at 2.5 μM significantly down-regulate several RP genes whereas OICR-9429 and MS67N did not have as significant of an effect at these lower concentrations (*Figure 3G*, *Figure 3—figure supplement 1E*). As down-regulation of RP genes expression implies a decrease in ribosome biogenesis, we also measured protein translation rates in TNBC cells where WDR5 was pharmacologically or genetically blocked. Accordingly, WDR5 silencing impaired global protein translation rates (*Figure 3H*, *Figure 3—figure supplement 1F*). OICR-9429 treatment or MS67-mediated degradation also caused decreases in protein translation in both LM2 and MDA-MB-453 cell lines (*Figure 3I and J*, *Figure 3—figure supplement 1G*). Taken together, our data demonstrates that either genetic or pharmacological inhibition of WDR5 can suppress RP gene expression and global translation in breast cancer cells.

## The WDR5-binding motif (WBM)-binding sites are required for WDR5-dependent cell growth and RP gene expression

We next sought to identify which of WDR5's multiple molecular function(s) is required for RP gene expression and breast cancer cell growth. WDR5 is canonically part of the mammalian KMT2A complex, which also consists of WRAD proteins (WDR5, RBBP5, ASH2L, and DPY30). Only KMT2A and RBBP5 interact directly with WDR5, and the complex has recently been elucidated by cryo-electron microscopy (*Park et al., 2019*; *Figure 4A*). WDR5 is a donut-shaped protein with two important binding pockets, WDR5-interacting (WIN) and WBM (*Guarnaccia and Tansey, 2018*; *Figure 4A and B*). KMT2A binding to the WIN site can be disrupted by point mutation F133A on WDR5 (*Guarnaccia et al., 2021*; *Patel et al., 2008*). On the other hand, RBBP5 and c-MYC have been shown to bind at the WBM site, which can be disrupted by point mutations N225A and V268E (*Guarnaccia et al., 2021*; *Thomas et al., 2015b*). In addition to the F133A, N225A, and V268E mutants, WDR5 mutants K7Q and 1-25Δ were recently shown to specifically impact ciliogenesis (*Kulkarni et al., 2018*).

Based on this information, we performed a structure function analysis of WDR5 by constitutively expressing shRNA-resistant wild-type (WT) WDR5 or the aforementioned WDR5 mutants with C-terminal 3XFlag-tag in LM2 cells, where endogenous *WDR5* was concomitantly reduced. Following DOX



**Figure 4.** The WBM-binding sites are required for WDR5-dependent cell growth and ribosomal protein gene expression. (**A**) WDR5 protein structure and key residues in the WIN and WBM sites that interact with binding partners. (**B**) Schematic of WDR5 with the indicated mutation sites. (**C**) Western blot analysis of the indicated proteins in LM2 inducible shWDR5 cells over-expressing wild-type (WT) WDR5 or WDR5 mutants. Cells were collected after 3 days of control or doxycycline (DOX) (1 μg/mL) induction. (**D**) Western blot analysis of the indicated proteins after immunoprecipitation using anti-Flag antibody in the LM2 shWDR5 cells over-expressing green fluorescent protein (GFP), WT WDR5, or WDR5 mutants. (**E–F**) Colony formation assays of cells expressing GFP, WT WDR5, or WDR5 mutants in inducible shControl (shCtrl) or shWDR5 cell lines after 9 days of control or DOX treatment. Representative images (**E**) and quantification (**F**) are shown. DOX-treated wells were compared to their respective controls for each cell line (n=3, unpaired two-side Student's $t$ test). $p_{WT}$ = 0.4047, ***$p_{GFP}$ = 0.0002, $p_{1-25\Delta}$=0.0854, *$p_{K7Q}$=0.0104, $p_{F133A}$=0.8448, **$p_{N225A}$=0.0014, and **$p_{V268E}$=0.0027. (**G**) RT-quantitative PCR analysis of the indicated mRNAs in LM2 from (**E**) induced with DOX (1 μg/mL) for 3 days. Significance determined by comparing each treatment to WT control (n=4, unpaired two-side Student's $t$ test). For *RPL9*, ****$p_{GFP}$ <0.0001, ****$p_{1-25\Delta}$<0.0001, ***$p_{K7Q}$=0.0002, ****$p_{N225A}$<0.0001, and ***$p_{V268E}$=0.0006; for *RPL31*, **$p_{GFP}$ = 0.0045, ***$p_{1-25\Delta}$=0.0007, **$p_{K7Q}$=0.0027, **$p_{N225A}$=0.0027, and *$p_{V268E}$=0.0307; for *RPS14*, ****$p_{GFP}$ <0.0001, ****$p_{1-25\Delta}$<0.0001, ****$p_{K7Q}$<0.0001, ****$p_{N225A}$<0.0001, and ***$p_{V268E}$=0.0005. For gel source data, see *Figure 4—source data 1* and *Figure 4—source data 2*.

The online version of this article includes the following source data and figure supplement(s) for figure 4:

**Source data 1.** Original western blots for *Figure 4C*.

**Source data 2.** Original western blots for *Figure 4D*.

**Figure supplement 1.** WDR5 recruitment to ribosomal protein gene promoters is not sufficient for gene activation.

induction, we confirmed ectopic WDR5 (mutant or WT) expression in the indicated mutant cell lines, whereas endogenous WDR5 levels were significantly repressed (*Figure 4C*). Using co-immunoprecipitation (Co-IP) assays, we observed that the F133A but not N25A or V268E mutations abrogate the binding of WDR5 to KMT2A. In contrast, mutants N225A and V268E but not F133A reduced the binding of WDR5 to RBBP5 by more than 50% as expected (*Figure 4D*). We next determined which WDR5 interacting site is required for cell growth. Consistently, shWDR5 cells expressing GFP control had severely impacted colony formation, while expression of WT WDR5 rescued this growth defect (*Figure 4E and F*). The N-terminal mutant 1-25Δ or K7Q has either a similar or slightly lower ability to rescue WDR5-dependent cell growth, respectively. Surprisingly, the WIN site mutant F133A was able to rescue the colony formation phenotype, while neither N225A nor V268E effectively rescued cell growth. These results suggest that the WBM but not the WIN-binding ability of WDR5 is required for WDR5-dependent growth of TNBC cells.

We next tested whether the different WDR5 mutants affected WDR5 binding to the promoter of RP genes and alter H3K4me3 levels in LM2 cells. Chromatin immunoprecipitation and qPCR (ChIP-qPCR) analysis showed that the F133A mutant binds to the promoter of *RPL7* and *RPL31* less efficiently, while N225A and V268E mutants bind chromatin similarly as the WT protein (*Figure 4—figure supplement 1A*). Surprisingly, all mutants maintained a similar level of H3K4me3 (*Figure 4—figure supplement 1B*), suggesting that WDR5 binding to chromatin is not required for maintaining H3K4me3 at the RP gene promoters tested in this context. More importantly, the N-terminal and F133A mutants rescued the expression of *RP* genes, whereas the N225A and V268E mutants did not (*Figure 4G*). Altogether, these data suggest that, in LM2 cells, WBM but not WIN binding by WDR5 is important for the maintenance of RP gene expression.

## Metastatic cell growth and lung colonization do not require the KMT2 complex components

The surprising observation that WIN binding by WDR5 is dispensable for breast cancer cell growth prompted us to directly test the requirement for the canonical KMT2 complex components in LM2 cells (*Figure 5A*). We first confirmed efficient knockdown of seven complex components (KMT2A, RBBP5, DPY30, HCFC1, CXXC1, WDR82, and BOD1L1), each with two independent shRNA after 3 days of DOX induction (*Figure 5B and C*). We first directly asked if KMT2A is required for LM2 cell growth and lung metastasis as KMT2A is a catalytic subunit of the H3K4 methyltransferase complex and was seemingly depleted in our screen (*Figure 1B and C*). However, depleting KMT2A with multiple shRNAs did not reproducibly affect in vitro colony formation and in vivo lung metastasis growth (*Figure 5D–F*). This result is also consistent with the phenotype observed from the F133A mutant and suggests that KMT2A is dispensable for WDR5-dependent cell growth. We next assessed whether RBBP5, DPY30, and HCFC1 are required for RP gene expression and cell growth. RP gene expression was not changed after knockdown of RBBP5, DPY30, or HCFC1 (*Figure 5G*). Furthermore, most of these KMT2 complex components are not required for the growth of LM2 cells (*Figure 5H and I*). The exception was upon knockdown of HCFC1 which resulted in a 40% decrease in colony formation (*Figure 5H and I*), likely due to the role of HCFC1 in cell cycle control (*Antonova et al., 2019*; *Xiang et al., 2020*). While RBBP5, DPY30, and HCFC1 are common to the KMT2 complexes, CXXC1 and WDR82 are distinct to the SET1A/B complexes, and BOD1L1 is specific to the SET1B complex (*Figure 5A*). Thus, we asked whether WDR5 regulates the growth phenotype specifically through SET1A/B complexes by perturbing CXXC1, WDR82, or BOD1L1. However, knockdown of these components also did not decrease RP gene expression and colony formation (*Figure 5G–I*). Therefore, the KMT2 complexes are not the major effectors of WDR5-dependent metastatic cell growth in the LM2 model.

## Inhibition of WDR5 and mTOR cooperatively reduces translation and TNBC growth

Hyperactivation of growth signaling pathways can increase protein synthesis, and inhibition of translation is being actively explored as a therapeutic avenue for cancer treatment (*Bhat et al., 2015*; *Grzmil and Hemmings, 2012*). Several mTOR inhibitors have been approved or are being tested in clinal trials, including the first generation mTOR inhibitors, everolimus and temsirolimus, and the second generation mTOR inhibitor, OSI-027 (*Zheng and Jiang, 2015*). Everolimus and temsirolimus



**Figure 5.** Metastatic cell growth and lung colonization do not require KMT2 complex components. (**A**) Schematic of subunit composition of several KMT2 complexes. (**B**) Western blot analyses of the indicated proteins in LM2 cells transduced with inducible shRNA targeting KMT2A, RBBP5, DPY30, HCFC1, CXXC1, and WDR82. Cells were collected after 3 days of doxycycline (DOX) (1 µg/mL) treatment. (**C**) RT-quantitative PCR (qPCR) analysis of *BOD1L1* in LM2 cells transduced with two independent hairpins targeting *BOD1L1*. Cells were collected after 3 days of DOX treatment (n=4, unpaired two-side Student's *t* test). (**D–E**) Colony formation assay of LM2 shCtrl or shKMT2A cells (shKMT2A-1 and shKMT2A-2) after 9 days of treatment with control or DOX. Representative images (**D**) and quantification (**E**) are shown (n=3, unpaired two-side Student's *t* test). (**F**) Normalized bioluminescence unit (BLU) signals of lung metastasis at day 66 of mice injected intravenously with LM2 cells from (**B**) and fed with DOX chow. The data represent mean ± SEM. Significance determined using unpaired two-tailed Mann–Whitney test. ns, not significant; p=0.2857 (shCtrl versus shKMT2A-1) and p=0.5556 (shCtrl versus shKMT2A-2). (**G**) RT-qPCR analysis of the noted RP genes in LM2 cells transduced with the indicated inducible shRNAs. Cells were collected after 3 days of DOX treatment. Significance determined by comparing each treatment to shCtrl. (**H–I**) Colony formation assay of LM2 cells

*Figure 5 continued on next page*

*Figure 5 continued*

from (**G**) after 9 days of either control or DOX treatment. Representative images (**H**) and quantification (**I**) are shown (n=3, unpaired two-side Student's *t* test). ***p=0.0005 (shHCFC1 #1 −DOX versus shHCFC1 #1 +DOX) and **p=0.0039 (shHCFC1 #2 −DOX versus shHCFC1 #2 +DOX). For gel source data, see ***Figure 5—source data 1***.

The online version of this article includes the following source data for figure 5:

**Source data 1.** Original western blots for ***Figure 5B***.

---

are rapalogs that allosterically inhibit mTORC1, while OSI-027 is an ATP-competitive inhibitor that inhibits both mTORC1 and mTOCR2 (***Zheng and Jiang, 2015***). Everolimus has been approved to treat post-menopausal women with advanced hormone receptor positive, HER2 negative breast cancer in combination with an aromatase inhibitor exemestane (***Baselga et al., 2012***). Because cancer cells could develop resistance to inhibitors of protein translation and this class of drugs may not directly cause tumor cell death (***Rozengurt et al., 2014***; ***Zheng and Jiang, 2015***), identifying other regimens which synergize with mTOR inhibitors is warranted.

Interestingly, during the course of titrating mTOR inhibitors in our cell model, we noted that the treatment with OSI-027 or everolimus alone caused an up-regulation of RP gene expression (***Figure 6—figure supplement 1A-D***), which may be due to an adaptive feedback effect on proteostasis following mTOR inhibition. Importantly, treatment with the WDR5 inhibitor OICR-9429 partially or completely blocked this adaptive induction of RP genes (***Figure 6—figure supplement 1A and B***). Moreover, while mTOR inhibitors were confirmed to down-regulate phosphorylated S6 protein kinase (S6K) and translation initiation factor 4E binding protein 1 (4E-BP1), WDR5 inhibition reduced RP genes expression and translation independently of this signaling pathway (***Figure 6A***, ***Figure 6—figure supplement 1E and F***).

Based on these results, we postulated that the inhibition of WDR5 and mTOR could cooperatively decrease TNBC protein translation, cell growth, and survival. As such we first treated LM2 and MDA-MB-453 cells for 3 days using OICR-4129 or the three mTOR inhibitors, everolimus, temsirolimus, and OSI-027. The levels of phosphorylated S6K and translation initiation factor 4E-BP1 were decreased in both cell lines after everolimus or temsirolimus treatment, while OSI-027 treatment only showed strong inhibition of mTOR signaling in the MDA-MB-453 cells (***Figure 6A and B***). Similar mTOR signaling inhibition was observed in LM2 cells expressing shWDR5, confirming that mTOR regulation is independent of WDR5 (***Figure 6—figure supplement 1F***). Next, we compared the global translation rates of LM2 or MDA-MB-453 cells, when WDR5 was inhibited genetically or pharmacologically in combination with mTOR inhibitors OSI-027 or everolimus. The overall translation was decreased when combining WDR5 inhibition or WDR5 knockdown with mTOR inhibition (***Figure 6C and D***, ***Figure 6—figure supplement 1G***). This combinatorial effect on protein translation correlated with an additive inhibition of clonogenic outgrowth (***Figure 6—figure supplement 1H–1K***). Moreover, both OSI-027 and everolimus act synergistically with WDR5 inhibition in both LM2 and MDA-MB-453 cells (***Figure 6E-L***), while temsirolimus showed an additive effect in LM2 cells (***Figure 6—figure supplement 1L and M***). We next tested the effects of MS67-mediated WDR5 degradation in combination with mTOR inhibition. MS67 treatment alone also did not affect mTOR signaling (***Figure 7A and B***). Interestingly, MS67 acts synergistically with both OSI-027 and everolimus in inhibiting translation in MDA-MB-453 cells (***Figure 7C***). Furthermore, we found that 5 µM MS67 is more effective than 20 µM OICR-9429 at inhibiting colony outgrowth when combined with either OSI-027 or everolimus in LM2 cells (compare ***Figure 6E–H*** to ***Figure 7D–G***). Importantly, we found that OSI-027 had better synergistic effects (based on coefficients of drug interactions) with WDR5 inhibition when compared to everolimus (***Figure 7D–G***), suggesting that mTORC2 could be critical for clonogenic outgrowth in the context of WDR5 inhibition. Moreover, we observed increased cleaved caspase 3 level in the combined treatment group in MDA-MB-453 cells (***Figure 6B***), suggesting the combination of WDR5 inhibition and OSI-027 induces apoptosis.

Collectively, our data identified WDR5-mediated protein translation as a potential vulnerability, which could be therapeutically leveraged in TNBC cells treated with first- or second-generation mTOR inhibitors.

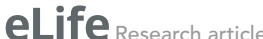

**Figure 6.** Inhibition of WDR5 and mTOR cooperatively reduces translation and triple-negative breast cancer (TNBC) cell growth. (**A**) Western blot analysis of the indicated proteins in LM2 cells treated for 3 days with or without 20 µM OICR-9429 in combination with control, 2 µM OSI-027, 2.5 µM temsirolimus, or 5 nM everolimus. (**B**) Western blot analysis of the indicated proteins in MDA-MB-453 treated for 3 days with or without 30 µM OICR-9429 in combination with control, 0.5 µM OSI-027, or 1 nM everolimus. (**C**) Normalized translational rates in LM2 cells from (**A**) (n=3, one sample *t* test). *p=0.0166 (DMSO versus OICR-9429), ***p=0.0001 (DMSO versus OSI-027), ** p=0.0099 (DMSO versus OICR-9429 +OSI-027), **p=0.0080 (DMSO versus everolimus), **p=0.0047 (DMSO versus OICR-9429+), *p=0.0232 (OSI-027 versus OICR-9429 +OSI-027), and *p=0.00352

*Figure 6 continued on next page*

Figure 6 continued

(everolimus versus OICR-9429 +everolimus). (**D**) Normalized translational rates in MDA-MB-453 cells from (**B**) (n=3, one sample *t* test). *p=0.0107 (DMSO versus OICR-9429), ****p<0.0001 (DMSO versus OSI-027), ****p<0.0001 (DMSO versus OICR-9429 +OSI-027), **p=0.0012 (DMSO versus everolimus), ***p=0.0005 (DMSO versus OICR-9429 +everolimus), ****p<0.0001 (OSI-027 versus OICR-9429 +OSI-027), and *p=0.0370 (everolimus versus OICR-9429 +everolimus). (**E–F**) Colony formation assay of LM2 treated for 8 days with or without 20 µM OICR-9429 in combination with control or 2 µM OSI-027. Representative images (**E**) and quantification (**F**) are shown. ****p<0.0001 (DMSO versus OICR-9429), ****p<0.0001 (DMSO versus OSI-027), ****p<0.0001 (DMSO versus OICR-9429 +OSI-027), ****p<0.0001 (OICR-9429 versus OICR-9429 +OSI-027), and *p=0.0233 (OSI-027 versus OICR-9429 +OSI-027). (**G–H**) Colony formation assay of LM2 cells treated for 8 days with or without 20 µM OICR-9429 in combination with control or 5 nM everolimus. Representative images (**G**) and quantification (**H**) are shown. **p=0.0043 (DMSO versus OICR-9429), **p=0.0011 (DMSO versus everolimus), ***p=0.0004 (DMSO versus OICR-9429 +everolimus), **p=0.0042 (OICR-9429 versus OICR-9429 +everolimus), and *p=0.0272 (everolimus versus OICR-9429 +everolimus). (**I–J**) Colony formation assay of MDA-MB-453 treated for 10 days with or without 30 µM OICR-9429 in combination with control or 0.5 µM OSI-027. Representative images (**I**) and quantification (**J**) are shown. **p=0.0018 (DMSO versus OICR-9429), ***p=0.0009 (DMSO versus OSI-027), ****p<0.0001 (DMSO versus OICR-9429 +OSI-027), ****p<0.0001 (OICR-9429 versus OICR-9429 +OSI-027), and *p=0.011 (OSI-027 versus OICR-9429 +OSI-027). (**K–L**) Colony formation assay of MDA-MB-453 cells treated for 10 days with or without 30 µM OICR-9429 in combination with control or 1 nM everolimus. Representative images (**K**) and quantification (**L**) are shown (n=3, unpaired two-side Student's *t* test). ***p=0.0002 (DMSO versus OICR-9429), ***p=0.0002 (DMSO versus everolimus), ****p<0.0001 (DMSO versus OICR-9429 +everolimus), ****p<0.0001 (OICR-9429 versus OICR-9429 +everolimus), and ***p=0.0003 (everolimus versus OICR-9429 +everolimus). Calculation of coefficients of drug interaction (CDIs) is described in Materials and methods section. Significant synergy is labeled with (#). For gel source data, see *Figure 6—source data 1* and *Figure 6—source data 2*.

The online version of this article includes the following source data and figure supplement(s) for figure 6:

**Source data 1.** Original western blots for *Figure 6A*.

**Source data 2.** Original western blots for *Figure 6B*.

**Figure supplement 1.** Inhibition of WDR5 and mTOR cooperatively reduces translation and triple-negative breast cancer (TNBC) growth.

**Figure supplement 1—source data 1.** Original western blots for *Figure 2—figure supplement 1E*.

**Figure supplement 1—source data 2.** Original western blots for *Figure 2—figure supplement 1F.*

## Discussion

Epigenetic aberrations contribute to multiple steps of tumor initiation, cancer invasion and migration, and tumor outgrowth through a wide spectrum of mechanisms (*Blair and Yan, 2012*; *Chen and Yan, 2021*). Moreover, recent efforts have led to the development of multiple pharmacological agents designed to target epigenetic and chromatin-modifying proteins in cancer (*Ahuja et al., 2016*; *Lu et al., 2020*; *Zhou et al., 2020*). Nevertheless, it is unclear how such agents can be leveraged therapeutically as single agents or in combination, particularly for the treatment of breast cancers. In this study, we performed an in vivo functional screen of epigenetic factors to identify WDR5 as being required for metastatic breast cancer growth. Intriguingly, WDR5 regulates ribosomal gene expression independent of its H3K4 methylation activity but through its WBM domain to mediate translation rate and cell growth. WDR5 inhibition or degradation suppresses translation and growth of breast cancer cells, alone or in combination with mTOR inhibitors. These results indicate that WDR5 promotes breast cancer growth and metastasis through regulating translation.

WDR5 is best known for its role in the KMT2 complexes, which promote transcription through H3K4 methylation (*Wysocka et al., 2005*). Unexpectedly, our structure function studies using the F133A WDR5 mutant suggest that KMT2 binding may not be critical for WDR5-mediated ribosomal gene expression and cell growth by metastatic TNBC cells. Consistently, depletion of several other components of the KMT2 complex did not affect the fitness of metastatic TNBC cells. These results suggest that WDR5 regulates translation and growth through KMT2 enzymatic activity-independent function. However, we showed that WIN site inhibitor OCIR-9429 suppresses RP gene expression and protein translation, suggesting that it exerts its effects through different mechanisms. It was reported that WIN site inhibitors C3 and C6 can displace WDR5 from the chromatin (*Aho et al., 2019*; *Bryan et al., 2020*), suggesting that OCIR-9429 may have similar activity. In addition, WIN site inhibitors may disrupt binding of proteins to the WBM site via allosteric effects, although C6 does not disrupt the binding between WDR5 and Myc or RBBP5 (*Guarnaccia et al., 2021*). These possibilities will need to be further investigated in future studies.

WDR5 can also be recruited to the NSL complex with the acetyltransferase male of the first (MOF), and WDR5 directly interacts with the subunit KANSL1 and KANSL2 through the WIN and WBM sites, respectively (*Dias et al., 2014*). The interaction of KANSL1 with WDR5 is important for efficient

**Figure 7.** MS67-mediated WDR5 degradation and mTOR inhibition cooperatively reduces translation and triple-negative breast cancer (TNBC) cell growth. (**A**) Western blot analysis of the indicated proteins in LM2 cells treated for 3 days with 2.5 µM MS67N or MS67 in combination control, 2 µM OSI-027, or 5 nM everolimus. (**B**) Western blot analysis of the indicated proteins in MDA-MB-453 cells treated for 3 days with 2.5 µM MS67N or MS67 in combination with control, 0.5 µM OSI-027, or 1 nM everolimus. (**C**) Normalized translational rates in MDA-MB-453 cells from (**B**) (n=3, one sample *t* test). *p=0.0209 (MS67N versus MS67), ****p<0.0001 (MS67N versus MS67N +OSI-027), ****p<0.0001 (MS67N versus MS67 +OSI-027), ****p<0.0001 (MS67N versus MS67N +everolimus), ****p<0.0001 (MS67N versus MS67 +everolimus), ****p<0.0001(MS67N +OSI-027 versus MS67 +OSI-027), and ****p<0.0001(MS67N +everolimus versus MS67 +everolimus). (**D–E**) Colony formation assay of LM2 cells treated for 9 days with 5 µM MS67N or MS67 in combination with control or 2 µM OSI-027. Representative images (**D**) and quantification (**E**) are shown. ****p<0.0001 (MS67N versus MS67), ****p<0.0001 (MS67N versus MS67N+OSI-027), ****p<0.0001 (MS67N versus MS67 +OSI-027), ***p=0.0004 (MS67 versus MS67 +OSI-027), and ****p<0.0001 (MS67N+OSI-027 versus MS67 +OSI027). (**F–G**) Colony formation assay of LM2 cells with 5 µM MS67N or MS67 in combination with control or 5 nM everolimus for 9 days. Representative images (**F**) and quantification (**G**) are shown. ***p=0.0005 (MS67N versus MS67), ***p=0.0004 (MS67N versus MS67N +everolimus), ***p=0.0002 (MS67N versus MS67 +everolimus), ***p=0.0003 (MS67 versus MS67 +everolimus), and ***p=0.0008 (MS67N +everolimus versus MS67 +everolimus). (n=3, unpaired two-side Student's *t* test). *p<0.05; **p<0.01; ***p<0.001; ****p<0.0001. Calculation of coefficients of drug interaction (CDIs) is described in Materials and methods section. Significant synergy is labeled with (#). For gel source data, see *Figure 7—source data 1* and *Figure 7—source data 2*.

*Figure 7 continued on next page*

*Figure 7 continued*

The online version of this article includes the following source data for figure 7:

**Source data 1.** Original western blots for *Figure 7A.*

**Source data 2.** Original western blots for *Figure 7B.*

targeting of NSL complex to the promoter of target genes (*Dias et al., 2014*). Therefore, it is likely that the NSL complex does not contribute to these phenotypes as WIN site WDR5 mutant F133A did not show a defective growth phenotype in this context. Alternatively, WDR5 likely regulates the phenotype described herein through a non-canonical function, such as its known ability to recruit the transcription factor MYC (*Thomas et al., 2015b*). WDR5 was previously shown to directly interact with MYC through the WBM site and facilitate the recruitment of MYC to chromatin (*Thomas et al., 2015b*). This is consistent with our findings that WBM site mutants of WDR5 are unable to rescue the growth defect caused by WDR5 loss. Notably, the association of MYC to its target genes is disrupted when the WBM site is mutated (*Thomas et al., 2015b*).

A recently published study implicates WDR5 in maintaining metastatic outgrowth via trimethylation of H3K4 on the promoters of specific target genes including *TGFB1*, which enhances epithelial to mesenchymal transition (EMT) (*Punzi et al., 2019*). Alternatively, by using a genome-wide approach and multiple TNBC cell line models, we did not observe alterations in EMT, which may be context specific. Conversely, we demonstrated a conserved and broad role for WDR5 in controlling RP gene expression (including *RPL32*, *RPL34*, *RPS14*, and *RPS6*) in a manner that is independent of KMT2 and H3K4me3 regulation at the promoters of RP genes. Therefore, the primary role of WDR5 may be to regulate proteostasis in TNBC cells. As aberrant protein translation affects multiple features of malignant cells, targeting WDR5 would be effective in treating both early or late stages of breast cancer (*Grzmil and Hemmings, 2012*). Consistent with this idea, knockdown of WDR5 independently decreases primary tumor growth and lung metastasis in vivo. Moreover, different RP genes and ribosomes may have specialized cellular functions (*Barna et al., 2022*). In fact, WDR5 knockdown also induces expression of several RP genes (including *RPL14*, *RPL21*, *RPS13*, and *RPS27*), which could have opposite roles as the RP genes down-regulated by WDR5 knockdown. Future studies will be needed to elucidate how WDR5-dependent protein translation contributes to the different steps of breast cancer progression, dissemination, and colonization.

The regulation of proteostasis and targeting protein translation in particular are potential therapeutic vulnerabilities of cancer cells. Interestingly, we demonstrated that the regulation of RP gene expression and protein translation could be inhibited by using the WDR5 inhibitor OICR-9429 or WDR5 degrader MS67, consistent with our genetic approach with WDR5 gene knockdown. Proteolysis-targeting chimeras (PROTACs) are hetero-bifunctional small molecules that can recruit desired target proteins to the E3 ubiquitin ligase complex for proteasomal degradation (*Paiva and Crews, 2019*). Multiple PROTAC degraders have entered clinical trials for cancer treatment (*He et al., 2020*). Here, we leveraged the newly designed WDR5 degrader to test its efficacy in WDR5 degradation in breast cancer cells. In fact, the WDR5 degrader MS67 showed more potent effects as compared to the WDR5 inhibitor OICR-9429. MS67 led to WDR5 degradation within 4 hr and is reversible after withdrawal of drug treatment (*Yu et al., 2021*), allowing for temporal control of WDR5 targeting. Unlike small molecule inhibitors, PROTAC molecules can be reused within the cells, which would lower the required concentration for drug treatment. Additionally, PROTAC is able to degrade the entire protein in the cells, which could overcome some potential drug resistant mechanisms. Our results thus suggest that WDR5 degradation is a potential therapeutic strategy to inhibit metastatic progression in breast cancer.

Finally, we discovered that multiple mTOR inhibitors can act synergistically with WDR5 targeting. In addition, we show that the second-generation mTOR inhibitor, OSI-027, which targets both mTORC1 and mTORC2, works better than first-generation inhibitor, everolimus, when treated in combination with WDR5 targeting. Both mTOR inhibition or WDR5 degradation can inhibit translation but through different mechanisms. mTOR integrates survival signals with protein synthesis. As translation initiation is initially repressed upon mTOR inhibition, negative feedback loops can cause aberrant stimulation of upstream signaling via AKT activation, which may diminish the effect of mTOR inhibitors (*Rozengurt et al., 2014*; *Zou et al., 2020*). We also observed up-regulation of RP gene expression after

mTOR inhibitor treatment, suggesting that epigenetic activation of ribosomal genes may be another compensatory response to mTOR inhibition. Importantly, we demonstrated that WDR5 inhibition is able to counteract this feedback activation of RP genes. Altogether, our study provides molecular and cell biological evidence that WDR5 is an important epigenetic mediator of protein translation and that this distinct function of WDR5 may be leveraged for treatment of TNBC.

# Materials and methods

**Key resources table**

| Reagent type (species) or resource | Designation | Source or reference | Identifiers | Additional information |
|---|---|---|---|---|
| Antibody | Anti-Flag (mouse monoclonal) | Sigma | F1804 | IP (1:50), WB (1:1000) |
| Antibody | Anti-vinculin (mouse monoclonal) | Sigma | V9131 | WB (1:10,000) |
| Antibody | Anti-tubulin (mouse monoclonal) | Sigma | T5168 | WB (1:10,000) |
| Antibody | Anti-WDR5 (rabbit monoclonal) | Cell Signaling Technology | #13105 | WB (1:1000) |
| Antibody | Anti-RBBP5 (rabbit monoclonal) | Cell Signaling Technology | #13171 | WB (1:1000) |
| Antibody | Anti-KMT2A/MLL1-C (rabbit monoclonal) | Cell Signaling Technology | #14197 | WB (1:1000) |
| Antibody | Anti-CXXC1 (rabbit polyclonal) | Cell Signaling Technology | #12585 | WB (1:1000) |
| Antibody | Anti-HCFC1 (rabbit polyclonal) | Cell Signaling Technology | #69690 | WB (1:1000) |
| Antibody | Anti-WDR82 (rabbit monoclonal) | Cell Signaling Technology | #99715 | WB (1:1000) |
| Antibody | Anti-phospho p70S6K (Thr389) (rabbit polyclonal) | Cell Signaling Technology | #9202 | WB (1:1000) |
| Antibody | Anti-phospho 4E-BP1 (Thr37/46) (rabbit monoclonal) | Cell Signaling Technology | #2855 | WB (1:1000) |
| Antibody | Anti-4E-BP1 (rabbit monoclonal) | Cell Signaling Technology | #9644 | WB (1:1000) |
| Antibody | Anti-cleaved caspase 3 (rabbit polyclonal) | Cell Signaling Technology | #9661 | WB (1:1000) |
| Antibody | Anti-DPY30 (rabbit polyclonal) | Bethyl Laboratories | A304-296A | WB (1:1000) |
| Antibody | Anti-phospho S6 (Ser240/244) (rabbit monoclonal) | Cell Signaling Technology | #5364 | WB (1:1000) |
| Antibody | Anti-S6 (rabbit monoclonal) | Cell Signaling Technology | #2217 | WB (1:10,000) |
| Antibody | Anti-M2 Flag (rabbit monoclonal) | Cell Signaling Technology | #14793 | Chromatin immunoprecipitation (ChIP) (10 µL) |
| Antibody | Anti-H3K4me3 (rabbit polyclonal) | Abcam | Ab8580 | ChIP (10 µg) |
| Recombinant DNA reagent | pINDUCER-10 (plasmid) | *Meerbrey et al., 2011* | | Inducible shRNA knockdown with red fluorescent protein reporter |
| Recombinant DNA reagent | pINDUCER-10-Blasticidin (plasmid) | This paper | | Version of pINDUCER-10 with blasticidin selection marker |
| Recombinant DNA reagent | p3XFlag-CMV-14-WDR5 | Addgene | #59974 | WDR5 plasmid used for subcloning |
| Recombinant DNA reagent | pDONR-211 | ThermoFisher Scientific | #12536017 | Used for cloning WDR5 into expression plasmid |
| Recombinant DNA reagent | pLenti-PSK-hygro-DEST | Addgene | #19066 | Used for cloning WDR5 into expression plasmid |
| Cell line (*H. sapiens*) | MDA-MB-231 | ATCC | HTB-26 | MDA-MB-231 expressing Luciferase |
| Cell line (*H. sapiens*) | MDA-MB-231-LM2 | Joan Massagué lab, Memorial Sloan Kettering Cancer Center | | Lung metastatic derivative of MDA-MB-231 |

*Continued on next page*

*Continued*

| Reagent type (species) or resource | Designation | Source or reference | Identifiers | Additional information |
|---|---|---|---|---|
| Cell line (*H. sapiens*) | MDA-MB-231-BoM | Joan Massagué lab, Memorial Sloan Kettering Cancer Center | | Bone metastatic derivative of MDA-MB-231 |
| Cell line (*H. sapiens*) | MDA-MB-231-BrM3 | This paper | | Brain metastatic derivative of MDA-MB-231 |
| Cell line (*H. sapiens*) | HCC1143 | ATCC | CRL-2321 | |
| Cell line (*H. sapiens*) | MDA-MB-453 | ATCC | HTB-131 | |
| Cell line (*H. sapiens*) | MCF7 | ATCC | HTB-22 | |
| Cell line (*H. sapiens*) | T47D | ATCC | HTB-133 | |
| Cell line (*H. sapiens*) | MDA-MB-361 | ATCC | HTB-27 | |
| Cell line (*H. sapiens*) | UACC893 | ATCC | CRL-1902 | |
| Cell line (*H. sapiens*) | BT474 | ATCC | HTB-20 | |
| Cell line (*H. sapiens*) | SKBR3 | ATCC | HTB-30 | |
| Cell line (*M. musculus*) | 4T1 | ATCC | CRL-2539 | |
| Chemical compound and drug | Puromycin | ThermoFischer Scientific | A1113802 | |
| Chemical compound and drug | Blasticidin | ThermoFischer Scientific | A1113902 | |
| Chemical compound and drug | Doxycycline | ThermoFischer Scientific | #446060050 | |
| Chemical compound and drug | MS67 | *Yu et al., 2021* | | WDR5 degrader |
| Chemical compound and drug | MS67N | *Yu et al., 2021* | | WDR5 degrader control |
| Chemical compound and drug | OICR-9429 | Sigma | SML1209 | |
| Chemical compound and drug | OSI-027 | Cayman Chemical | #17379 | |
| Chemical compound and drug | Everolimus | Cayman Chemical | #11597 | |
| Chemical compound and drug | Temsirolimus | Cayman Chemical | #11590 | |
| Software and algorithm | R studio | R studio | RRID:SCR_000432 | |
| Software and algorithm | STAR | STAR | RRID:SCR_004463 | |
| Software and algorithm | GENCODEv96 | GENCODEv96 | RRID:SCR_014966 | |
| Software and algorithm | PyMol | PyMol | RRID:SCR_000305 | |
| Commercial assay and kit | Cell proliferation reagent WST-1 | Roche | #11644807001 | |
| Commercial assay and kit | TruSeq Stranded mRNA Library Prep | Illumina | #20020594 | |
| Commercial assay and kit | High-capacity cDNA Reverse Transcription Kit | ThermoFisher | #4368813 | |
| Commercial assay and kit | Click-iT Cell Reaction Buffer Kit | ThermoFisher | C10269 | |

## Antibodies and chemicals

For Co-IP and western blots, the following antibodies were obtained commercially: mouse anti-Flag (M2, F1804), mouse anti-vinculin (V9131), and mouse anti-tubulin (T5168) (Sigma, St. Louis, MO); rabbit anti-WDR5 (#13105), rabbit anti-RBBP5 (#13171), rabbit anti-KMT2A/MLL1-C (#14197), rabbit anti-CXXC1(#12585), rabbit anti-HCFC1 (#69690), rabbit anti-WDR82 (#99715), rabbit anti-phospho p70 S6K (Thr389) (#9205), rabbit anti-p70 S6K (#9202), rabbit anti-phospho 4E-BP1 (Thr37/46) (#2855), rabbit anti-4E-BP1 (#9644), and anti-cleaved caspase 3 (#9661) (Cell Signaling Technology, Danvers, MA, USA); rabbit anti-DPY30 (A304-96A) (Bethyl Laboratories, Montgomery, TX, USA).

For drug treatment experiments, WDR5 inhibitor OICR-9429 (Sigma, SML1209 and Cayman Chemical, #16095) and control OICR-0547 (Structural Genomics Consortium), and mTOR inhibitors, OSI-027 (Cayman Chemical, #17379), everolimus (Cayman Chemical, #11597), and temsirolimus (Cayman Chemical, #11590) were used. Compounds of WDR5 degrader MS67 and negative control MS67N were synthesized in Jian Jin's lab.

## Plasmids and virus generation

Frozen bacterial stocks harboring the shRNA library were generated by the Westbrook lab. pGIPZ plasmid harboring hairpins and barcodes were digested with Xho I and Mlu I and subcloned into the pINDUCER10 plasmid. The list of hairpin sequences and shRNA knockdown efficiency is available in *Supplementary file 1a*. For cloning of the WDR5 mutants, BP (atttB x attP) cloning primers were designed against p3XFlag-CMV-14-WDR5. Two-step PCR was performed to generate shRNA resistant mutant WDR5. Briefly, two sets of primers were designed such that they overlap at the site of mutagenesis. The product from the PCR was then used for BP (Thermo Fisher, # 11789020) or LR (attL x attR, Thermo Fisher, #11791020) reaction into pDONR-211 or pLenti-PSK-hygro-DEST. p3XFlag-CMV-14-WDR5 was a gift from Debu Chakravarti (Addgene, #59974). A list of cloning oligos is available in *Supplementary file 1e*.

For virus generation, HEK293T cells were transfected with 1.2 µg each of VSV-G, TAT, RAII, and HyPM packaging plasmids along with 11.2 µg of lentiviral plasmid. OptiMEM and TransIT-293 Transfection Reagent (Mirus, MIR2700) were used following manufacturer protocol. Viruses were collected at 48 hr and 72 hr and filtered through a 0.45 µm filter.

## Cell culture and stable cell lines generation

MDA-MB-231 and its metastatic derivatives, MDA-MB-231-LM2 (LM2), MDA-MB-231-BoM (BoM) and MDA-MB-231-BrM3 (BrM3) breast cancer cells and HEK293T cells were cultured in Dulbecco's Modified Eagle Medium supplemented with 10% fetal bovine serum and 100 U/mL penicillin and 100 µg/mL streptomycin. LM2 and BoM have been previously described and were obtained from J. Massagué (Memorial Sloan Kettering Cancer Center, New York) (*Minn et al., 2005*; *Kang et al., 2003*). BrM3 is a metastatic derivative generated by subjecting MDA-MB-231-BrM2 cells to one round of in vivo selection (*Bos et al., 2009*). HCC1143, MDA-MB-453, MCF7, T47D, MDA-MB-361, UACC893, BT474, SKBR3, and 4T1 breast cancer cells were cultured in RPMI1640 supplemented with 10% fetal bovine serum, 100 U/mL penicillin, and 100 µg/mL streptomycin. Cells were periodically tested for mycoplasma contamination and were negative, and they were authenticated using short tandem repeat profiling with GenePrint 10 system (Promega, #B9510).

For generation of cell lines, viruses harboring pINDUCER10-puromycin or pINDUCER10-blasticidin constructs were titrated using the target cell lines. Cells were infected at an multiplicity of infection of 1 and selected using either 0.8 µg/mL puromycin or 10 µg/mL blasticidin. For generation of cell lines harboring WDR5 mutants, optimal viral dose was determined empirically by western blot visualization to assess equal expression of WDR5 across mutant cell lines. LM2 cells with re-introduction of WDR5 mutants were selected with 800 µg/mL hygromycin.

## Minipool generation for in vitro and in vivo screening

Minipools were created by equally mixing 8–10 individual LM2 cell lines harboring pINDUCER10 hairpins targeting each epigenetic modifier together with two LM2 positive control cell lines (shBUD31 and shSAE2) and two negative control cell lines (shCHEK1 and shSTAMBP). For in vitro screening, minipool cells were plated into 10 cm dishes with or without 1 µg/mL of DOX. A portion of minipool cells were collected as day 0 samples as the controls. Every 2 days, the cells were pelleted, and all

samples were proceeded to gDNA isolation and gDNA qPCR. For in vivo screening, $5×10^5$ minipool cells were injected into nude mice through tail vein. Lung metastases were monitored weekly with in vivo live imaging. At the end point, the mice were sacrificed, and the lung tissue was harvested for gDNA isolation and gDNA qPCR. For the screening readout analyses, all qPCR results were normalized to the value from day 0. The fold change was obtained from +DOX/−DOX for both in vitro and in vivo screen. A table of complete results of both in vivo and in vitro screens is available in *Supplementary file 1b*.

## Animal studies

Female athymic Nude-*Foxn1$^{nu}$* immunodeficient (6–8 weeks old) mice (Envigo) were used for lung-metastasis experiments with human cell lines. For in vivo screening, $5×10^5$ cells were injected via tail vein in 0.1 mL saline. For WDR5 in vivo validation experiment, cells were treated with DOX for 3 days prior to injection, and $2×10^5$ cells were injected via tail vein in 0.1 mL saline. Mice were placed on DOX chow (Envigo, TD.01306) 5 days prior to injection. All the in vivo metastasis signals, including lung metastasis and whole-body metastasis, were monitored by weekly bioluminescence imaging with an IVIS system coupled to Living Image acquisition and analysis software (Xenogen). Luminescence signals were quantified at the indicated time points as previously described. Values of luminescence photon flux of each time point were normalized to the value obtained immediately after xenografting (day 0).

For mammary fat pad tumor assays, control and shWDR5-1 LM2 cells ($1×10^6$) were resuspended in 0.1 mL of saline and matrigel (Corning, #356231) mix and then injected into mammary fat pad (the fourth mammary glands) of non-obese diabetic-severe combined immunodeficiency (NOD-SCID) mice (6 weeks old). Tumor were monitored every 7 days by measuring the tumor length (L) and width (W). Tumor volume was calculated as $V=L×W^2/2$. Mice were euthanized when primary tumors reached 1000 mm$^3$. All animal procedures have been approved by the Institutional Animal Care and Use Committee of Yale University under animal protocol 2021–11286.

## Lung tissue harvest and gDNA isolation

Mice were sacrificed and whole body perfused with 10 mL of PBS. For gDNA isolation, the harvested lungs were placed into a microcentrifuge tube and snap frozen with liquid nitrogen. The frozen tissues were then placed into an aluminum block on dry ice. Each tube of the lung tissue was allowed to thaw enough for further mincing with surgical scissors and then refrozen by dipping them in liquid nitrogen bath. This process was repeated two to three times until no visible tissue chunk was observed. 60 mg of homogenized tissue was then aliquoted out and processed with the QIAmp DNA mini kit (Qiagen 51304) following manufacturer's protocols.

## Western blot and Co-IP

Cells were lysed in 1× high salt lysis buffer (50 mM Tris-HCl pH 8, 320 mM NaCl, 0.1 mM EDTA, 0.5% NP-40, and 10% glycerol) or [radioimmunoprecipitation assay (RIPA) buffer (50 mM Tris-HCl pH 7.4, 150 mM NaCl, 1 mM EDTA, 1% Triton X-100, 1% sodium deoxycholate, and 0.1% sodium dodecyl sulfate (SDS)) supplemented with 1× protease inhibitor (Roche cOmplete 11836153001). Cell lysates were vortexed and centrifuged, and the supernatants were subjected to protein quantification by Bradford reagent (Bio-Rad 5000006) and sample preparation by sample buffer (10% glycerol, 50 mM Tris-HCl [pH 6.8], 2% SDS, 0.01% bromophenol blue, and 8% β-mercaptoethanol). Protein samples were resolved by SDS-PAGE according to the standard protocol and transferred onto 0.45 μm nitro-cellulose membranes (Bio-Rad 1620115) and blotted with the primary and secondary antibodies as described.

For Co-IP experiments, cells were lysed with RIPA buffer. The prepared protein extracts were precleared with protein A/G beads (Pierce, #20421) for 1 hr at 4°C then incubated with anti-Flag M2 affinity gel for 2 hr for Co-IP, followed by western blot analysis.

## Colony formation assays and WST-1 cell proliferation assays

Colony formation assays were done by seeding single cells in 6 or 12 well plates. Media was replenished every 3 days with indicated treatments. Colonies were fixed in 4% para-formaldehyde (PFA), followed by 0.5% crystal violet staining for 30 min at room temperature and rinsed with water. Quantification

was performed using the ImageJ software plugin ColonyArea. Statistical significance was determined using unpaired, two-tailed Student's $t$ test performed on intensity values from ColonyArea. For WST-1 cell proliferation assays (#11644807001, Roche), cells were seeded in 96 well plate for indicated days growth and then were assayed according to the manufacturer's instructions.

## RNA-sequencing

Cells from knockdown control or shWDR5-1 group were harvested with QIAzol Lysis Reagent (Qiagen) and homogenized using QIAshredder tubes (Qiagen). For each cell line, shRNA expression was induced with DOX (1 μg/mL) for 3 days, and three biological replicates were harvested at different passages. RNA isolation was performed using miRNeasy with on-column DNase digestion. External RNA controls consortium spike-in RNA was added in proportion to the number of cells obtained during cell counts. Library generation was performed using TruSeq stranded mRNA library prep kit (Illumina). Paired-end sequencing was performed using an Illumina HiSeq4000 sequencer, generating an average of 59 million reads per library. Reads were aligned to hg38, and gene counts to GENCODEv96 transcripts were obtained using STAR aligner v2.7.0 with default parameters. The hg38 and GENCODEv96 annotations were appended to include the ERCC sequences. DESeq2 was used to obtain differential gene expression, and HTSFilter was used to filter for expressed genes. Significant differences were identified using a Benjamini-Hochberg adjusted p-value cut-off of 0.05. RNA-seq data have been deposited into the National Center for Biotechnology Information Gene Expression Omnibus database under GSE196666.

## RT-qPCR and barcode qPCR

Total RNA was extracted using the RNeasy Plus Mini Kit (Qiagen 74136), and reverse transcription was performed using High-capacity cDNA Reverse Transcription kit (ThermoFisher 4385614). The resulting cDNA was diluted with water, and Fast SYBR Green Master Mix (ThermoFisher 4385614) was used for real-time PCR. *GAPDH* was utilized as loading controls. Samples were run in quadruplicate, and the experiments were performed at least three times. Primer sequences are listed in *Supplementary file 1f*.

For barcode qPCR, barcode primers were designed to amplify only 1 barcode sequence among the 89 unique barcodes in the entire library. The primer set targeting the tetracycline response element (TRE) element in pINDUCER10 was used for normalization. The full list of barcode qPCR primers used for detection of hairpin abundance is available in *Supplementary file 1g*.

## Translation rate assay

Cells were starved of L-methionine for 30 min and subsequently incubated with 50 μM homopropargylglycine (HPG; Life Technologies #C10186) for 1–4 hr in treatment media. Cells were then trypsinized and fixed in 4% PFA. A Click-IT kit (Life Technologies #C10269) used to label HPG. Labeled cells were analyzed using an Cytoflex flow cytometer. Translation rates were determined based on the slope of HPG incorporation over time. Significance determined using one sample $t$ test to compare each treatment value to the hypothetical value 1 for the DMSO control.

## Chromatin immunoprecipitation qPCR

Cells growth in 15 cm dishes were washed with PBS and cross-linked with 1% formaldehyde in DMEM media for 10 min and quenched with 0.125 M glycine for 5 min. Cells were washed with cold PBS and scraped and pooled. Following washes, cell pellets were lysed in sonication buffer (20 mM Tris pH 8.0, 2 mM EDTA, 0.5 mM EGTA, 1× protease inhibitor, 0.5% SDS, and 0.5 mM phenylmethanesulfonyl fluoride (PMSF)) at a concentration of 3 mL per $1 \times 10^8$ cells for 10 min. Sonication was performed using the Qsonica Q800R sonicator (Qsonica) set to 70% amplitude, 15 s on and 45 s off for a total of 30 min on. Sonicated materials were precleared with 50% protein A agarose (ThermoFisher 20421). Antibodies were added into precleared material and rotated overnight at 4°C. 50% protein A slurry was then added, and the tubes were rotated at 4°C for 2 hr. In order to reverse crosslinks and purify DNA, NaCl was added to elute ChIP material and incubated overnight at 65°C and then digested with proteinase K. Glycotube (ThermoFisher AM9515) was added as co-precipitant and phenol-chloroform isolation, and ethanol precipitation was performed to isolate ChIP DNA. All sample DNA pellets were

resuspended in 200 µL of water. 2 µL of DNA was used for each qPCR reaction, and reactions were performed in quadruplicate.

## 3D protein visualization

Protein crystal structure 2H14 (apo-WDR5) was downloaded from the Protein Data Bank and visualized using PyMol (The PyMOL Molecular Graphics System, Version 2.0 Schrdinger, LLC).

## Analysis of in vitro drug interaction

We employed coefficient of drug interaction to determine cytotoxicity. The coefficient of drug interaction (CDI) is calculated as follows: CDI = AB/(A×B). According to the colony formation intensity or translation rates of each group, AB is the ratio of the combination group to the control group; A or B is the ratio of the single agent group to the control group. Thus, CDI value <1, =1, or >1 indicates that the drugs are synergistic, additive, or antagonistic, respectively. A CDI <0.7 indicates a significant synergistic effect (*Otahal et al., 2020*; *Zhao et al., 2014*).

## Statistical analysis

Comparisons between two groups were performed using an unpaired two-side Student's *t* test unless indicated otherwise. Graphs represent either group mean values ± SEM or individual values (as indicated in the figure legends). For animal experiments, each tumor graft was an independent sample. All experiments were reproduced at least three times.

## Acknowledgements

We would like to thank all members of Yan, Nguyen, and Stern laboratories at Yale University for helpful discussions, Dr. Mei Zhong at Yale Stem Cell Center Genomics Core facility for helping with sample preparation for RNA-seq, Dr. Joan Massagué at Memorial Sloan Kettering Cancer Center for providing MDA-MB-231, LM2, and BoM cells, Dr. Yibin Kang at Princeton University for providing BrM2 cells, Dr. Narendra Wajapeyee at the University of Alabama Birmingham for helping with compiling the epigenetic gene list. Sequencing done at Yale Stem Cell Center Genomics Core facility was supported by the Connecticut Regenerative Medicine Research Fund and the Li Ka Shing Foundation. Figure panels 1 A, 4B, 5 A, and Figure 1—figure supplement 1A were created with BioRender.com. The control drug OICR-0547 was supplied by the Structural Genomics Consortium under an Open Science Trust Agreement: http://www.thesgc.org/click-trust.

## Additional information

### Competing interests

Jian Jin: The Jin laboratory received research funds un-related to this study from Celgene Corporation, Levo Therapeutics, Inc, Cullgen, Inc and Cullinan Oncology, Inc. J.J. is a cofounder, scientific advisory board member and equity shareholder in Cullgen, Inc and a consultant for Cullgen, Inc, EpiCypher, Inc and Accent Therapeutics, Inc. Don X Nguyen: has received research funding un-related to this study from AstraZeneca Inc. The other authors declare that no competing interests exist.

### Funding

| Funder | Grant reference number | Author |
|---|---|---|
| National Science Foundation | Graduate Research FellowshipDGE-1122492 | Wesley L Cai |
| National Cancer Institute | F31CA243295 | Jocelyn Fang-Yi Chen |
| Congressionally Directed Medical Research Programs | W81XWH-15-1-0117 and W81XWH-21-1-0411 | Qin Yan |
| National Cancer Institute | R01CA237586 | Qin Yan |

| Funder | Grant reference number | Author |
|---|---|---|
| National Cancer Institute | R01CA166376 | Don X Nguyen |
| National Cancer Institute | P30CA016359 | Don X Nguyen<br>Qin Yan |
| Yale Cancer Center | Class of '61 Cancer Research Award | Don X Nguyen<br>Qin Yan |
| National Institutes of Health | 1S10OD025132 and 1S10OD028504 | Jian Jin |

The funders had no role in study design, data collection and interpretation, or the decision to submit the work for publication.

## Author contributions

Wesley L Cai, Conceptualization, Resources, Data curation, Formal analysis, Funding acquisition, Validation, Investigation, Methodology, Writing – original draft, Project administration, Writing – review and editing; Jocelyn Fang-Yi Chen, Conceptualization, Data curation, Formal analysis, Funding acquisition, Validation, Investigation, Visualization, Methodology, Writing – original draft, Project administration, Writing – review and editing; Huacui Chen, Emily Wingrove, Meiling Zhang, Minghui Zhao, Amer Balabaki, Wenxue Li, Ethan D Krop, Investigation; Sarah J Kurley, Methodology; Lok Hei Chan, Yansheng Liu, Investigation, Methodology; Anna Arnal-Estape, Investigation, Writing – review and editing; Xufen Yu, Jian Jin, Thomas F Westbrook, Resources, Methodology; Yali Dou, Resources; Don X Nguyen, Conceptualization, Supervision, Funding acquisition, Investigation, Methodology, Project administration, Writing – review and editing; Qin Yan, Conceptualization, Supervision, Funding acquisition, Investigation, Writing – original draft, Project administration, Writing – review and editing

## Author ORCIDs

Wesley L Cai http://orcid.org/0000-0002-3871-5509
Jocelyn Fang-Yi Chen http://orcid.org/0000-0001-7281-8686
Anna Arnal-Estape http://orcid.org/0000-0003-0490-7040
Minghui Zhao http://orcid.org/0000-0001-8928-9258
Amer Balabaki http://orcid.org/0000-0002-1509-886X
Xufen Yu http://orcid.org/0000-0001-7794-7890
Yansheng Liu http://orcid.org/0000-0002-2626-3912
Jian Jin http://orcid.org/0000-0002-2387-3862
Don X Nguyen http://orcid.org/0000-0003-0324-5604
Qin Yan http://orcid.org/0000-0003-4077-453X

## Ethics

All animal procedures have been approved by the Institutional Animal Care and Use Committee of Yale University under animal protocol 2021-11286.

## Decision letter and Author response

Decision letter https://doi.org/10.7554/eLife.78163.sa1
Author response https://doi.org/10.7554/eLife.78163.sa2

---

# Additional files

### Supplementary files

• Supplementary file 1. Information related to shRNA characterization, in vivo and in vitro screening, RNA-seq analysis, cloning oligos and primers. (a) List of hairpin sequences and knockdown efficiency. (b) Results of in vivo and in vitro screen. l2fc, Log2 fold change (c) Shared differentially expressed genes (DEGs) after WDR5 knockdown in LM2, BrM3, and BoM. l2fc, Log2 fold change; padj, adjusted p value. (d) Enriched pathways of differentially expressed genes (DEGs) after WDR5 knockdown in LM2, BrM3, BoM, or all three cell lines (common). Enriched pathways, significance, combined score, and overlapping genes are shown. (e) List of cloning oligos. (f) List of primers used for RT-qPCR. (g) List of barcode qPCR primers.

• Transparent reporting form

## Data availability

RNA-seq data have been deposited into the National Center for Biotechnology Information (NCBI) Gene Expression Omnibus database under GSE196666.

The following dataset was generated:

| Author(s) | Year | Dataset title | Dataset URL | Database and Identifier |
|---|---|---|---|---|
| Cai WL, Chen JF, Yan Q | 2022 | WDR5 promotes breast cancer growth and metastasis via KMT2-independent translation regulation | https://www.ncbi.nlm.nih.gov/geo/query/acc.cgi?acc=GSE196666 | NCBI Gene Expression Omnibus, GSE196666 |

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
