## [Editor Report]

Using combined in vivo and in vitro screens, this study identifies WDR5 as important for tumour growth and lung metastasis in a triple negative breast cancer. WDR5 promotes global translation rates and enhanced ribosomal protein expression, and targeting of WDR5 in combination with MTOR inhibition effectively reduces tumour cell growth and metastasis. Novel therapeutic strategies for triple negative breast cancer are urgently needed, and this study elegantly provides a novel therapeutic strategy that may contribute to improved clinical management of this patient population.

---

## [Decision Letter]

**Decision letter after peer review:**

Thank you for submitting your article "WDR5 promotes breast cancer growth and metastasis via KMT2-independent translation regulation" for consideration by *eLife*. Your article has been reviewed by 2 peer reviewers, and the evaluation has been overseen by a Reviewing Editor and Richard White as the Senior Editor. The following individuals involved in review of your submission have agreed to reveal their identity: Greg Wang (Reviewer #1) and David Heery (Reviewer #2).

Essential revisions:

1) 1/OICR-9429 is believed to target the WDR5-KMT2 interaction; meanwhile, the structure function study shown in Figure 4 and the genetic study in Figure 5 showed that WDR5's activation effects on ribosomal protein (RP) genes are independent of KMT2. These data almost indicate that OICR-9429 might affect WBM mediated WDR5 interaction to partners such as MYC, maybe via allosteric effect? Please explain these findings and expand the possible implications of these results more.

2) Initially it was stated that the focussed library covers 100 targets, but later it is stated that 7 minipools were used to cover the 69 targets in both screens? A list of the non-discovery target screened might also be useful in supplementary data, e.g. did these include other components of KMT2 or NSL complexes? please explain.

*Reviewer #1 (Recommendations for the authors):*

Overall, this study is well designed, comprehensive and well done. Also, the manuscript is generally well written and the data is well presented as well. The findings reported here in breast cancer can potentially be broadly applicable to many other cancers. The finding that the regulation of ribosome genes by WDR5 is independent of KMT2 is also novel. I would therefore support the publication at *eLife* after the below issues are addressed.

1/OICR-9429 is believed to target the WDR5-KMT2 interaction; meanwhile, the structure function study shown in Figure 4 and the genetic study in Figure 5 showed that WDR5's activation effects on ribosomal protein (RP) genes are independent of KMT2. These data almost indicate that OICR-9429 might affect WBM mediated WDR5 interaction to partners such as MYC, maybe via allosteric effect? Can author explain and expand a bit more?

2/ KMT2-WDR5 complex accounts for H3K4me1/2/3. And authors checked H3K4me3 in their functional studies in Figure 4-supplement. Would be better to test the H3K4me2 level in the ChIP-qPCR experiments.

*Reviewer #2 (Recommendations for the authors):*

Overall a very impressive and well-controlled study.

---

## [Author Response]

Essential revisions:1) 1/OICR-9429 is believed to target the WDR5-KMT2 interaction; meanwhile, the structure function study shown in Figure 4 and the genetic study in Figure 5 showed that WDR5's activation effects on ribosomal protein (RP) genes are independent of KMT2. These data almost indicate that OICR-9429 might affect WBM mediated WDR5 interaction to partners such as MYC, maybe via allosteric effect? Please explain these findings and expand the possible implications of these results more.

We appreciate the reviewers’ prediction and suggestion. We examined WDR5 binding to various interacting proteins when cells were treated with two “WDR5-KMT2” (WIN site) inhibitors: MM-401 or OICR-9429. This experiment shows that the interaction between WDR5 and c-Myc is indeed decreased by MM-401 and OICR-9429, likely via an allosteric effect (Author response image 1). Both inhibitors also disrupt RBBP5 or ASH2L interactions, albeit to a lesser extent (Author response image 1). Furthermore, it is reported that WIN site inhibitors C3 and C6 can displace WDR5 from the chromatin (Aho et al., 2019 Cell Rep PMID: 30865883 and Bryan et al. 2020 NAR, PMID: 31996893), consistent with our data that OCIR-9429 suppresses RP gene expression. However, C6 does not disrupt the binding between WDR5 and c-Myc or RBBP5 (Guarnaccia et al. 2021 Cell Rep, PMID: 33472061), suggesting that these different WIN site inhibitors exert different mechanisms of action. Therefore, we mention these possibilities as avenues for future studies with appropriate references in the revised Discussion.

**Author response image 1. sa2fig1:** c-Myc and WDR5 interaction is disrupted by WDR5 inhibitors MM-401 or OICR9429. (A) Western blot of M2-Flag or IgG control IP in Flag-WDR5-expressing LM2 cells after treatment with the indicated concentrations of MM-401. DMSO was used as treatment control. (B) Western blot of M2-Flag or IgG control IP in Flag-WDR5-expressing LM2 cells after treatment with inactive control compound OICR-0547 or the indicated concentrations of OICR-9429. DMSO was used as treatment control.

2) Initially it was stated that the focussed library covers 100 targets, but later it is stated that 7 minipools were used to cover the 69 targets in both screens? A list of the non-discovery target screened might also be useful in supplementary data, e.g. did these include other components of KMT2 or NSL complexes? please explain.

Thank you for pointing out the confusing description of the library that was assembled built and screened. Because some pINDUCER10 shRNA constructs did not efficiently knockdown our targets, we only screened part of the library that we initially assembled. We have now added the knockdown efficiency of shRNA in pGIPZ in the new Table S1 and indicated the shRNAs that have been used in in vitro and in vivo screens in Table S1. We have also added a new Table S2, which summarizes the results of the discovery and non-discovery targets in both in vivo and in vitro conditions. We have clarified the text to state the procedures more clearly and updated the numbers.

Of note, we were focused on screening for “druggable” epigenetic targets, and therefore we did not include additional components of the NSL complexes. Likewise, our library included only two other components of the KMT2 complex: SETD1A, which is a non-discovery target, and KMT2A, which is a depleted target. However, as we have described in the initial submission, knockdown of KMT2A with multiple shRNAs did not reproducibly affect in vitro colony formation and in vivo lung metastasis growth (Figure 5D-F)”.

Reviewer #1 (Recommendations for the authors):Overall, this study is well designed, comprehensive and well done. Also, the manuscript is generally well written and the data is well presented as well. The findings reported here in breast cancer can potentially be broadly applicable to many other cancers. The finding that the regulation of ribosome genes by WDR5 is independent of KMT2 is also novel. I would therefore support the publication at eLife after the below issues are addressed.1/OICR-9429 is believed to target the WDR5-KMT2 interaction; meanwhile, the structure function study shown in Figure 4 and the genetic study in Figure 5 showed that WDR5's activation effects on ribosomal protein (RP) genes are independent of KMT2. These data almost indicate that OICR-9429 might affect WBM mediated WDR5 interaction to partners such as MYC, maybe via allosteric effect? Can author explain and expand a bit more?

Please refer see our reply to comment (1) under “Essential Revisions” above.

2/ KMT2-WDR5 complex accounts for H3K4me1/2/3. And authors checked H3K4me3 in their functional studies in Figure 4-supplement. Would be better to test the H3K4me2 level in the ChIP-qPCR experiments.

We have not profiled H3K4me2 in that setting, which in our experience usually does not change as an intermediate histone mark. Our H3K4me3 results are consistent with a previous study (Aho et al., 2019 Cell Rep PMID: 30865883) that showed transcription changes caused by WIN inhibitors are not dependent on H3K4me3 alterations.